# Amplicon Sequencing Analysis of Submerged Plant Microbiome Diversity and Screening for ACC Deaminase Production by Microbes

**DOI:** 10.3390/ijms252413330

**Published:** 2024-12-12

**Authors:** Binoop Mohan, Aqsa Majeed, Doni Thingujam, Sethson Silton Burton, Katie E. Cowart, Karolina M. Pajerowska-Mukhtar, M. Shahid Mukhtar

**Affiliations:** 1Department of Biology, University of Alabama at Birmingham, 3100 East Science Hall, 902 14th Street South, Birmingham, AL 35294, USA; binoopm2@uab.edu (B.M.); amajeed@clemson.edu (A.M.); dthingu@clemson.edu (D.T.); ssburton@uab.edu (S.S.B.); kecowart@uab.edu (K.E.C.); kmukhta@clemson.edu (K.M.P.-M.); 2Biosystems Research Complex, Department of Genetics & Biochemistry, Clemson University, 105 Collings St., Clemson, SC 29634, USA; 3Department of Biological Sciences, Clemson University, 132 Long Hall, Clemson, SC 29634, USA

**Keywords:** aquatic plants, stress tolerance, co-occurrence network analysis, natural sources

## Abstract

Submerged plants can thrive entirely underwater, playing a crucial role in maintaining water quality, supporting aquatic organisms, and enhancing sediment stability. However, they face multiple challenges, including reduced light availability, fluctuating water conditions, and limited nutrient access. Despite these stresses, submerged plants demonstrate remarkable resilience through physiological and biochemical adaptations. Additionally, their interactions with microbial communities are increasingly recognized as pivotal in mitigating these environmental stresses. Understanding the diversity of these microbial communities is crucial for comprehending the complex interactions between submerged plants and their environments. This research aims to identify and screen microbes from submerged plant samples capable of producing 1-aminocyclopropane-1-carboxylic acid (ACC) deaminase and to explore microbial diversity through metagenomic analysis. Microbes were isolated and screened for ACC deaminase production, and metagenomic techniques, including co-occurrence network analysis, were used to examine microbial diversity and interactions within the communities. ACC deaminase-producing microbes can significantly enhance plant metabolism under stress conditions. The identification of the culturable bacteria revealed that most of these microbes belong to the genera *Pseudomonas*, *Bacillus*, and *Acinetobacter*. A total of 177 microbial strains were cultured, with molecular identification revealing 79 reductant, 86 non-reductant, and 12 uncultured strains. Among 162 samples screened for ACC deaminase activity, 50 tested positive. To further understand microbial dynamics, samples were collected from both natural sources and artificial pond reservoirs to assess the impact of the location on flood-associated microbiomes in submerged plants. Metagenomic analysis was conducted on both the epiphytic and endophytic samples. By exploring the overall composition and dynamics of microbial communities associated with submerged plants, this research seeks to deepen our understanding of plant–microbe interactions in aquatic environments. The microbial screening helped to identify the diverse microbes associated with ACC deaminase activity in submerged plants and amplicon sequencing analysis paved the way towards identifying the impact of the location in shaping the microbiome and the diversity associated with endophytic and epiphytic microbes. Co-occurrence network analysis further highlighted the intricate interactions within these microbial communities. Notably, ACC deaminase activity was observed in plant-associated microbes across different locations, with distinct variations between epiphytic and endophytic populations as identified through co-occurrence network analysis.

## 1. Introduction

Submerged aquatic vegetation (SAV) comprises plants that grow fully underwater. These plants are vital to aquatic ecosystems, providing habitats and sustenance for diverse aquatic organisms, enhancing water quality through oxygenation, and stabilizing sediments. SAV is crucial for maintaining an ecological equilibrium in both freshwater and marine environments. Aquatic ecosystems are increasingly at risk due to the combined impacts of flooding and climate change [1]. Flooding events can lead to hypoxic conditions, while climate change can alter water temperatures and nutrient availability. Aquatic ecosystems face numerous critical threats that threaten plant microbiomes and the overall health of the ecosystem. Flooding and extreme weather events introduce sediment, debris, and pollutants into aquatic ecosystems, potentially suffocating aquatic life. Nutrient pollution from agricultural runoff and sewage can cause eutrophication and hypoxia, while heavy metals and organic pollutants can accumulate and become toxic. Habitat destruction due to development and the introduction of invasive species can disrupt food webs and decrease biodiversity. Climate change is worsening many of these issues, causing shifts in species distributions and the collapse of sensitive environments. Addressing these complex risks is essential to protect the health of aquatic ecosystems and the microbial communities they sustain [2,3]. These changes have profound implications for submerged plants, which play a vital role in maintaining water quality and ecosystem stability [4]. Moreover, the microbiome associated with submerged plants significantly contributes to their ability to adapt and thrive under these challenging environmental conditions. The unique characteristics of these microbial communities offer promising opportunities for enhancing plant stress tolerance and promoting sustainable agriculture and environmental conservation [5,6].

Studies on submerged plant microbiomes have revealed intricate interactions between plants and their microbial associates, shaped by factors such as salinity, nutrient levels, and the temperature, all of which influence plant stress resilience. Understanding the specific roles of individual microbial species within these communities provides valuable insights into enhancing the overall resilience of submerged plants. For instance, microbial interactions within the plant holobiont are crucial for maintaining plant health and stress tolerance. Furthermore, there is significant interest in the genetic and metabolic capabilities of submerged plant microbiomes [7]. Metagenomic analyses have uncovered a diverse array of genes and biochemical pathways within these microbial communities, highlighting their potential to facilitate nutrient cycling, form biofilms, and produce metabolites that enhance plant stress tolerance [8,9]. Additionally, the communication between submerged plant hosts and their microbiomes involves complex signaling mechanisms that regulate the plant’s stress response. This intricate network of chemical signaling and molecular communication underscores the sophisticated nature of these plant–microbe interactions [10].

As our understanding of submerged plant microbiomes improves, their potential applications in sustainable agriculture and environmental management become increasingly evident. Research has shown that microbial diversity in wetlands significantly affects the biogeochemical processes associated with nutrient cycling. Harnessing the beneficial characteristics of these microbial communities can enhance crop resilience, reduce reliance on synthetic inputs, and mitigate the environmental impact of agricultural practices. Exploring submerged plant microbiomes holds promise for developing innovative solutions to address challenges in food security and ecosystem sustainability [11]. Submerged plant microbiomes, encompassing diverse microbial communities inhabiting the rhizosphere and surrounding environments of submerged aquatic plants, play pivotal roles in the ecology and physiology of these plants. Among the myriad microbial functions, the production of ACC deaminase by certain microbial species within these communities stands out as a key mechanism influencing plant growth and stress tolerance [12]. Flooded plants generally have limited gas exchange and hence the accumulation of ethylene will trigger a plant stress response. ACC deaminase is an enzyme capable of cleaving ACC, a precursor of ethylene, into ammonia and α-ketobutyrate [13,14]. By reducing the levels of ACC and subsequently ethylene, ACC deaminase-producing microbes alleviate the inhibitory effects of ethylene on plant growth, particularly under stress conditions.

Research on the microbiomes associated with submerged aquatic plants is crucial for tackling significant environmental challenges. By deepening our understanding of the diversity and functions of these microbial communities, we can discover innovative applications for practical benefits. In agriculture, utilizing beneficial microbes, such as those producing ACC deaminase, could promote more sustainable and climate-resilient cropping systems. For ecological restoration, knowledge of how microbiomes support aquatic ecosystem health can inform efforts to rehabilitate degraded habitats. From an environmental management standpoint, studying the complex microbial interactions that drive nutrient cycling and ecosystem functions can shape policies to protect sensitive aquatic environments. Overall, this research on submerged plant microbiomes is vital for advancing sustainable agricultural practices, restoring degraded ecosystems, and enhancing environmental restoration efforts.

The current studies on submerged plant microbiomes do not provide a clear understanding of the diversity of microbes. This study primarily focused on exploring the role of native submerged plants including *Myriophyllum spicatum*, *Najas guadalupensis*, *Heteranthera dubia*, and *Ceratophyllum demersum*, which were used to understand the role of the microbiome in submerged plants. Our work aimed to explore microbial diversity through co-occurrence network analysis to estimate the dynamics of an active community. The work will enable us to understand the significance of the submerged plant microbiome in facilitating plant adaptations to environmental stresses, with a particular focus on the role of ACC deaminase production in improving plant resilience and productivity [15]. In this study, we aimed to identify culturable microbes and perform molecular identification, followed by screening for ACC deaminase-positive strains. The amplicon sequencing analysis characterized the complexity of the network system and the kinetics of its interactions and provided a far deeper insight into the diversity of bacteria connected to the submerged plants than the co-expression network analysis alone.

## 2. Results

### 2.1. Submerged Plant Sampling and ACC Deaminase Screening

Samples of submerged plants were collected from fifteen locations near the University of Alabama at Birmingham. The specific locations and their GPS (Global Positioning System) coordinates are provided in Appendix A. Please modify the sentence as “Plant samples from these locations were surface-sterilized to remove surface microbes. The homogenized, isolated endophytic microbes were initially spread-plated, and individual colonies were streak-plated on Luria-Bertani (LB) media. The isolated microbes were screened in DF (Dworkin and Foster) + ACC media and a total of 177 endophytic microbial strains were isolated, and molecular identification was performed on these samples. Detailed information about the microbial strains is presented in Appendix A. Out of these, 162 microbial strains were screened from fifteen geographically distinct locations. Among these, 50 strains demonstrated ACC deaminase activity. These ACC deaminase-positive microbial strains were further characterized by their colony-forming unit (CFU) counts through serial dilutions to assess growth and viability. Microbes that demonstrated significant growth in DF + ACC media compared to minimal growth in DF media were identified as ACC deaminase-positive strains. The CFU counts exhibited significant variation among the positive strains, reflecting a diverse range of ACC deaminase activity levels. The dilutions at which ACC deaminase production was observed for each microbial strain are summarized in Appendix A. The CFU counts at different dilutions highlight the diverse metabolic capabilities of these strains. For instance, strains such as A11, A20, and H3 demonstrated robust growth and viability, as indicated by their high dilution levels (6). In contrast, strains like B2, B8, and F9 exhibited relatively lower growth rates, as suggested by their lower dilution levels (1). The positive strains were distributed across nine locations, with the highest number of positive strains (12) isolated from Location 3, followed by Location 5 with 10 positive strains and Location 7 with 8 positive strains. The remaining locations had between three and seven positive strains each. This distribution pattern indicates a widespread occurrence of ACC deaminase-producing microbes in various environments. To understand the microbial diversity, a metagenomic approach was carried out on samples collected from a natural source and an artificial pond (Appendix A). Both epiphytic and endophytic samples from the plant samples were used in the experiment to further elucidate the composition of the microbial community that exists in a submerged plant microbiome.

### 2.2. Characterization of Microbiome Diversity Through Taxonomic Annotation

The plant samples collected from location A and H were *Myriophyllum spicatum* and *Najas minor*, respectively. The isolation of the DNA for amplicon sequencing analysis was performed from the shoot samples collected from location A and H, and DNA isolation was performed in both epiphytic and endophytic samples. We analyzed the 16S region of the amplicon sequencing samples and its taxonomic analysis revealed a diverse bacterial community within the plant flood microbiome. Figure 1a illustrates the distribution of three dominant phyla, *Proteobacteria*, *Firmicutes*, and *Bacteroidota*. Additionally, smaller quantities of *Actinobacteriota*, *Acidobacteriota*, and *Chloroflexi* were observed. Intriguingly, the *Bdellovibrionota* phylum, known for its predatory behavior towards other bacteria, was the least abundant.

We further performed taxonomic annotations at the family level of the plant flood microbiome, comparing endophytic and epiphytic samples from two different locations as shown in Figure 1b. The endophytic samples, EnA3 and EnH3, displayed a diverse distribution of microbial families, with a significant presence of *Sphingomonadaceae* and *Moraxellaceae*. In contrast, the epiphytic samples, EpA3 and EpH3, exhibited a different distribution, with a higher presence of *Rhodocyclaceae*, *Pseudomonadaceae*, and *Oxalobacteraceae*. The ‘Others’ category in both sample types represented various other microbial families present at lower frequencies. These data provide valuable insights into the microbial distribution in plant flood microbiomes and highlight the variations between endophytic and epiphytic samples from different locations.

To explore in depth, we also carried out the taxonomic annotations at the genus level of the plant flood microbiome, comparing endophytic and epiphytic samples from two different locations (Figure 1c). The endophytic samples, EnA3 and EnH3, showed a diverse distribution of microbial genera, with a significant presence of *Duganella* and *Massilia*. The ‘Others’ category in these samples represented a large portion, indicating the presence of various other microbial genera at lower abundances. Conversely, the epiphytic samples, EpA3 and EpH3, exhibited a different distribution, with a higher frequency of *Janthinobacterium*, *Pseudomonas*, and *Flavobacterium*. The ‘Others’ category was less abundant in these samples, indicating a lesser diversity of microbial genera compared to the endophytic samples. These data provide valuable insights into the microbial distribution in plant flood microbiomes at the genus level and how it varies between endophytic and epiphytic samples from different locations. The presence of the variations among *the microbial* genera in the endophytic and epiphytic samples suggests that the living conditions (endophytic or epiphytic) and location can significantly affect the microbial composition in plant flood environments.

### 2.3. Comparison of Differential Abundance Analysis 

To understand the differences among the microbial communities, a linear discriminant analysis (LDA) score plot, a statistical tool, was used to differentiate and visualize variations in microbial communities between samples (Figure 2a). In this study, the samples were categorized as endophytic (EnA3 and EnH3) and epiphytic (EpA3 and EpH3), each from distinct locations. Each bar in the plot represents a specific bacterial taxonomic classification at various levels, with the length of the bar indicating the LDA score, which reflects the degree of difference in the abundance of that bacterial classification between the groups. The endophytic and epiphytic samples exhibited distinct distributions of microbial taxa, suggesting that both the type of sample and the location significantly influenced the microbial composition in plant flood environments. Certain bacterial classifications were more prevalent in specific samples, with the genera *Duganella* and *Massilia* being more abundant in endophytic samples, and *Janthinobacterium*, *Pseudomonas*, and *Flavobacterium* being more abundant in epiphytic samples. The ‘Others’ category included various other microbial taxa present in smaller amounts. This result provides valuable insights into the microbial distribution in plant flood microbiomes at the genus level, highlighting the variations between endophytic and epiphytic samples from different locations. The roles of these taxa in the plant flood microbiome may include nutrient cycling, disease resistance, and adaptation to flooding conditions.

We also tried to explore the potential role of a bacterial order being abundant in the samples under investigation. In the plot, each bacterial order is listed on the *y*-axis, and their corresponding relative abundance in each group is shown on the *x*-axis (Figure 2b). The length of each bar represents the proportion of that bacterial order within the sample group. The plot reveals that certain bacterial orders, such as *Burkholderiales*, *Rhizobiales*, and *Sphingomonadales*, have a higher relative abundance across all sample types. Conversely, orders like *Myxococcales* and *Xanthomonadales* have a lower relative abundance or are absent in some groups. Comparing endophytic samples (EnA3 and EnH3) with epiphytic samples (EpA3 and EpH3) shows differences in the relative abundance of these bacterial orders, indicating that the type of sample (endophytic or epiphytic) and the location can significantly influence the microbial composition in plant flood environments.

It is vital to understand the interaction kinetics among the various samples and to understand the overlapping communities. The Venn diagram was constructed to illustrate the intricate interplay of microbial communities in plant flood microbiomes sampled from different locations in Figure 2c. The unique microbiomes found in each sample were 200 for EnA3, 118 for EnH3, 52 for EpA3, and 37 for EpH3. The number of shared microbiomes between EnA3 and EnH3 was 25, between EnA3 and EpA3 was 2, between EnA3 and EpH3 was 10, between EnH3 and EpA3 was 16, between EnH3 and EpH3 was 28, and between EpA3 and EpH3 was 24. There were 26 microbes shared among all four sample sets. Other overlaps of note included five between EnA3 and EpH3, eight between EnH3 and EpA3, and thirteen between EnH3 and EpH3. This finding underscores the complexity of these microbial communities and their potential role in plant adaptations to flood conditions.

### 2.4. Beta and Alpha Diversity

A Kruskal–Wallis Rank Sum Test revealed substantial differences in alpha diversity among the studied locations (*p* = 0.015). The Shannon index, which considers both the species abundance and evenness, varied among locations [Figure 3a], with EnH3 exhibiting the highest median value, indicating a more balanced and diverse ecosystem. The Chao1 index, estimating the species richness, also differed among locations, with EnA3 having the highest median value, suggesting a greater species richness. Evenness scores, reflecting the distribution of species, demonstrated that EnH3 had the highest median value, indicating a more even species distribution. These results highlight the substantial impact of environmental conditions on the community composition and biodiversity, as evidenced by variations in the species richness, evenness, and entropy-based indices. To understand beta diversity [Figure 3b], a PERMANOVA analysis using a reduced model revealed significant differences among the locations (*p* = 0.002). The distinct clustering observed in the PCoA (Principal Coordinate Analysis) plot indicates that microbial community compositions vary significantly between locations, likely influenced by site-specific environmental factors.

### 2.5. Co-Occurrence Network Analysis Among Bacterial Taxa

The correlation patterns between bacterial taxa were analyzed using four co-occurrence networks constructed based on Spearman’s correlation coefficient (r > 0.6, *p* < 0.05) (Figure 4). The clustering coefficient represents the network’s complexity and the strength of interactions among microorganisms. The average clustering coefficient (avgCC) and modularity were higher in the four networks compared to random networks (Table 1. Previous studies have shown that a higher clustering coefficient corresponds to a more dynamic and active community within the network [16]. The numbers of nodes and edges were highest in EpA3, followed by EnH3, EnA3, and EpH3 (Table 1). Additionally, the EpA3 and EnH3 networks exhibited higher average degrees and modularity than the EnA3 and EpH3 networks. In the EpA3 network, there were 502 positive and 440 negative edges. The negative correlation ratio was lower than the positive correlation ratio in EpA3 compared to the other three networks.

## 3. Discussion

Our study offers a comprehensive analysis of the microbial diversity associated with submerged plants, focusing on ACC deaminase-producing microbes. Additionally, there was a significant variation in ACC deaminase activity levels among the isolated microbial strains. While some strains showed strong growth and viability in the presence of ACC, others had noticeably lower activity [17,18]. This finding is consistent with earlier research, which indicates that the variability in ACC deaminase activity among different strains can be attributed to factors such as their taxonomic classification, metabolic pathways, and environmental conditions [19,20]. Further investigation into these factors could provide valuable insights into how these microbes enhance plant resilience to flooding stress.

The identification of 50 microbial strains capable of producing ACC deaminase in submerged plant microbiomes shows great potential for improving plant stress tolerance and promoting sustainable agriculture. These microbes utilize a key mechanism by breaking down ACC (a precursor of ethylene) into α-ketobutyrate and ammonia, which helps lower ethylene levels in plants experiencing flooding stress [21]. This reduction in ethylene is crucial for maintaining normal root development, enhancing water and nutrient uptake, and boosting the overall plant resilience under waterlogged conditions [18,22]. The high prevalence of these beneficial microbes in our samples indicates a strong potential for developing targeted bioinoculants to help crops better withstand flooding stress. This approach is particularly relevant given the increasing frequency of extreme weather events due to climate change, offering a sustainable method to enhance agricultural productivity in flood-prone areas while reducing reliance on chemical inputs.

The taxonomic analysis identified a diverse bacterial community, with *Proteobacteria*, *Firmicutes*, and *Bacteroidota* as the dominant phyla. These phyla are vital to the breakdown of organic materials because they lower nitrite and nitrate levels, recycle nutrients, and aid in energy metabolism—all processes that are necessary for submerged plants to survive and grow [23]. The flood microbiome analysis also revealed a diverse bacterial community, predominantly comprising *Proteobacteria*, *Firmicutes*, and *Bacteroidota*. These findings align with previous research that has identified these phyla as common in various environmental microbiomes [24,25].

*Proteobacteria*, known for their metabolic diversity and adaptability, are vital in nutrient cycling, including nitrogen fixation and organic compound degradation [26,27]. Their prevalence in the flood microbiome suggests a significant contribution to ecosystem functionality, particularly in nutrient cycling and organic matter decomposition. *Firmicutes*, recognized for their ability to form endospores, demonstrate resilience in harsh conditions and are associated with the initial stages of organic matter decomposition, particularly cellulose degradation [28,29]. Their abundance in the flood microbiome indicates their role in breaking down plant material and other organic substrates. *Bacteroidota*, efficient in degrading complex carbohydrates, are crucial in the later stages of organic matter decomposition, complementing the activities of *Firmicutes* [24,25]. The interaction between these dominant phyla highlights the complexity and functionality of the flood microbiome. *Proteobacteria* likely initiate the breakdown of organic matter, creating conditions favorable for *Firmicutes* and *Bacteroidota* to further decompose complex substrates. This sequential degradation process ensures efficient nutrient cycling and organic matter turnover in flood environments.

Additionally, we observed less abundant phyla such as *Actinobacteriota*, *Acidobacteriota*, and *Chloroflexi*. These phyla likely contribute to specific functions, including the production of antifungal compounds and antibiotics that improve plants’ response towards pathogens [6]. The presence of the *Bdellovibrionota* phylum, known for its predatory behavior towards other bacteria, was intriguing and warrants further investigation. The linear discriminant analysis (LDA) score plot and the Venn diagram provided further insights into the intricate interplay of microbial communities in plant flood microbiomes. The variations in the microbial composition between endophytic and epiphytic samples from different locations underscore the influence of environmental factors on the microbial diversity of submerged plants [7]. A significant observation was the difference in microbial diversity between endophytic and epiphytic samples, as well as across various sampling sites. The taxonomic analysis at the family and genus levels (Figure 1b,c) highlighted distinct microbial community compositions. Endophytic samples showed a higher overall diversity, with the notable representation of *Sphingomonadaceae* and *Moraxellaceae*. In contrast, epiphytic samples were dominated by *Rhodocyclaceae*, *Pseudomonadaceae*, and *Oxalobacteraceae*. These findings are consistent with previous studies, which have indicated that epiphytic communities are generally more diverse and less stable than endophytic ones [30]. Moreover, the plant identity has been shown to have a more pronounced impact on the diversity and structure of epiphytic bacteria compared to endophytic bacteria [31]. Additionally, research on the grapevine leaf microbiome has revealed that epiphytic isolates are phylogenetically more diverse than endophytic isolates [32]. 

## 4. Materials and Methods

### 4.1. Submerged Plant Sampling and Processing

Submerged plants were collected from 15 locations within the Greater Birmingham Area in Alabama, USA, as listed in Appendix A. This study primarily focused on exploring the role of submerged plants found in Alabama including *Myriophyllum spicatum*, *Najas guadalupensis*, *Heteranthera dubia*, and *Ceratophyllum demersum*, which were used in our experiment to screen microbes. Samples, which included both the plants and their associated water, were gathered using sterile 50 mL tubes. Sterile tweezers and scissors were used for collection, and each sample was labeled with details such as the location and plant type. The samples were transported in ice bags and stored in a laboratory refrigerator at 4 °C until processed. A small portion of the collected sample was used for processing. The samples were cut using sterile scissors and transferred to a 50 mL sterile tube, then covered with an autoclaved net cloth. A soap solution was added, and the sample was gently shaken for 60 s. The soap solution was then rinsed off and the sample was washed multiple times with running water. Next, a 70% ethanol solution was added and mixed for 40 s before being drained. A 2% bleach solution was then added, mixed for 1 min and 40 s, and subsequently rinsed off. Finally, the samples were washed with sterile distilled water. Using sterile tweezers, the samples were transferred to a 2 mL microcentrifuge tube containing grinding beads and 1 mL of 1X PBS buffer. The samples were homogenized at a speed of 6 m/s for 40 s.

### 4.2. Media Preparation and Plating

To prepare LB agar solid media for microbial screening, 25 g of LB broth and 15 g of agar were added to 1 L of sterile distilled water. The mixture was gently stirred using a magnetic stirrer to prevent clumping. The prepared medium was then autoclaved at 121 °C and 20 psi for 30 min. After autoclaving, the medium was allowed to cool on a magnetic stirrer. Once it reached 55 °C, the medium was transferred to a laminar flow hood and poured into 100 × 15 mm Petri plates. The plates were left to solidify and then labeled. For spread plating, 100 µL of the homogenized sample was used. The plates were sealed with parafilm and incubated overnight at 28 °C. Individual microbial colonies from the overnight spread plate culture were then used for streak plating on LB media. Sterile loops were used to carefully pick single colonies from the spread plate, and quadrant plating was performed to isolate individual microbes from the culture.

### 4.3. Screening for ACC Deaminase-Producing Microbes

A DF medium, as described by [33], was used to screen for microbes capable of producing ACC deaminase. Three different formulations of DF media were utilized in this screening step. The first medium contained only DF media without any nitrogen source, the second medium included DF media supplemented with (NH_4_)_2_SO_4_, and the third medium had (NH_4_)_2_SO_4_ replaced with 3 mM ACC. One liter of each medium was prepared and autoclaved at 121 °C for 20 min. Individual cultures were added to a 96-well plate containing 1X PBS solution, followed by the serial dilution of the samples. The samples were then plated onto the respective screening media and incubated at 28 °C for 1–2 days. The growth patterns of microbes in each medium were observed to identify ACC deaminase-producing microbes.

### 4.4. DNA Isolation and Amplification of 16S rDNA

Microbial cultures were incubated overnight in LB broth media. A 10 µL aliquot of the culture was transferred to a 200 µL PCR tube. An alkaline lysis buffer, consisting of 100 mM sodium hydroxide and 100 mM disodium EDTA, was used to lyse the microbial sample. Specifically, 16.6 µL of the lysis buffer was added to the culture. The mixture was gently agitated using a 200 µL pipette and then incubated at 95 °C for 30 min. After incubation, the reaction mixture was allowed to cool to room temperature, and 16.6 µL of a neutralization buffer containing Tris-HCl was added. The molecular identification of the microbes was performed using universal primers: the forward primer AGAGTTTGATCCTGGCTCAG and the reverse primer ACGGCTACCTTGTTACGACTT [34]. The PCR conditions for amplifying the 16S rDNA region included an initial denaturation at 95 °C for 5 min, followed by 30 cycles of denaturation at 95 °C for 30 s, annealing at 55 °C for 30 s, and elongation at 72 °C for 2 min. A final elongation step was carried out at 72 °C for 10 min, followed by holding at 4 °C.

### 4.5. DNA Isolation and Amplicon Sequencing Library Preparation

For DNA amplification, we targeted the V4 region of the 16S rRNA gene sequence. The forward primer sequence was 515F-Y (5′-GTGYCAGCMGCCGCGGTAA-3′) paired with the reverse primer 806R (5′-GGACTACNVGGGTWTCTAAT-3′). The amplification process utilized Phusion High-Fidelity DNA Polymerase following the protocol provided by the supplier. DNA isolation from epiphytic and endophytic plant samples was conducted using the Qiagen DNeasy PowerSoil Pro Kit, following the manufacturer’s instructions. All steps related to library preparation, quality control testing, and the sequencing of the amplicon sequencing samples were performed at BGI in Cambridge, MA, USA.

### 4.6. Data Analysis

The amplicon sequencing data analysis was carried out using 3 bioreps for each sample studied, and the variability in the study is represented in Appendix A. The demultiplexed paired raw sequences were quality-filtered and denoised using the DADA2 algorithm within the Qiime2 pipeline [35]. Amplicon sequence variants (ASVs) were aligned with 99% similarity to the Silva 138 reference database [36]. Microbial diversity analyses, including alpha and beta diversities, Principal Coordinate Analysis (PCoA), and PERMANOVA, were conducted using the vegan package [37] in RStudio (v.4.4.0). The Bray–Curtis distance matrix was utilized for PCoA analysis. The top 10 most abundant bacterial phyla, families, and genera were visualized in bar plots using the plot_bar function in the microeco package [38].

### 4.7. Co-Occurrence Network Analysis

For co-occurrence network analysis, samples were divided into four groups, resulting in different numbers of OTUs in each group (Table 1). Only ASVs (amplicon sequence variants) present in at least 50% of samples and with a relative abundance greater than 0.1% were included in the analysis. Spearman’s correlations with r > 0.6 and *p* < 0.05 were used to construct the network. Network properties, including the average path length, average degree, clustering coefficient, and modularity, were analyzed using the ggClusterNet package [39]. Key characteristics are summarized in Table 1.

## 5. Conclusions

Our study sheds light on the intricate and diverse microbial communities associated with submerged plants. The screening process identified 50 microbial strains with notable ACC deaminase activity, demonstrating their potential for biotechnological applications. The variation in CFU counts at different dilutions underscores the diverse metabolic capabilities of these strains, which could be utilized for various agricultural and environmental purposes. We emphasized the crucial role of ACC deaminase-producing microbes in enhancing plant resilience and productivity. Our findings highlight the potential of leveraging these microbial communities to improve crop resilience, reduce reliance on synthetic inputs, and mitigate the environmental impact of agricultural practices. Future research should focus on elucidating the specific roles of less abundant microbial taxa and their interactions with dominant phyla. Additionally, exploring the role of predatory bacteria in these communities presents an exciting avenue for future investigation. Our study paves the way for harnessing the power of submerged plant microbiomes to address challenges in food security and ecosystem sustainability.

## Figures and Tables

**Figure 1 ijms-25-13330-f001:**
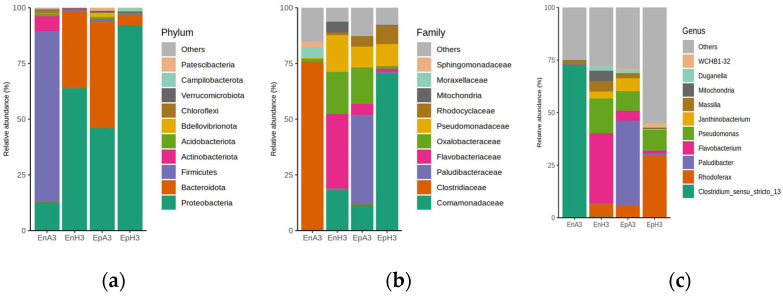
Bar graphs illustrating the bacterial community compositions at different taxonomic levels across four samples (Endophytic A3 (EnA3), Endophytic H3 (EnH3), Epiphytic A3 (EpA3), Epiphytic H3 (EpH3)) at (**a**) the phylum level, (**b**) family level, and (**c**) genus level, depicting the shifts in microbial diversity and abundance.

**Figure 2 ijms-25-13330-f002:**
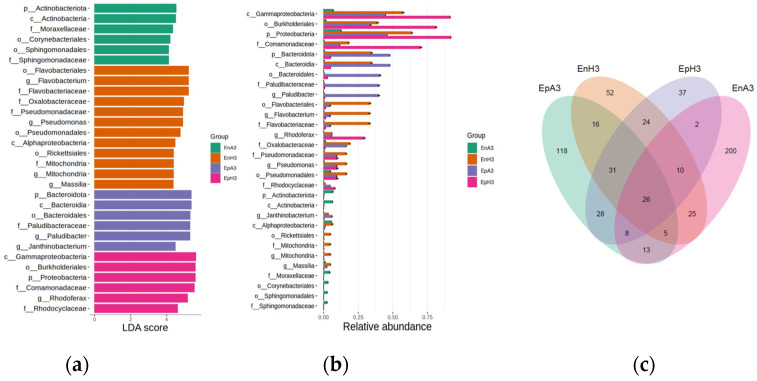
(**a**) Bar chart illustrating the distribution of various species; the *x*-axis represents linear discriminant analysis (LDA) scores, indicating the discriminatory power of each species between the groups. Distinct colors represent diverse groups. (**b**) Horizontal bar chart comparing the relative abundance of species across three groups. The length of each bar represents the species’ abundance, with segments of distinct colors indicating the proportion of each species within each group. (**c**) Venn diagram depicting the overlap between four datasets labeled EnA3, EnH3, EpA3, and EpH3. The overlapping regions represent shared data points, with numerical values indicating the quantity of shared elements.

**Figure 3 ijms-25-13330-f003:**
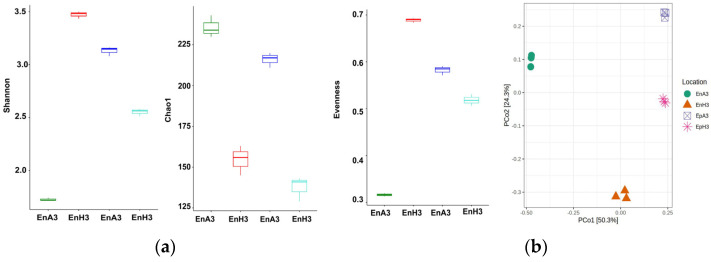
(**a**) Alpha diversity metric distribution: This plot illustrates the distribution of alpha diversity metrics among various sample groups. Each group is represented by a unique color and label on the *x*-axis, while the *y*-axis quantifies the alpha diversity metric. (**b**) Beta diversity PCoA scatter plot: this plot presents a two-dimensional ordination of samples based on beta diversity measures, providing a visual representation of the similarity or dissimilarity between microbial communities.

**Figure 4 ijms-25-13330-f004:**
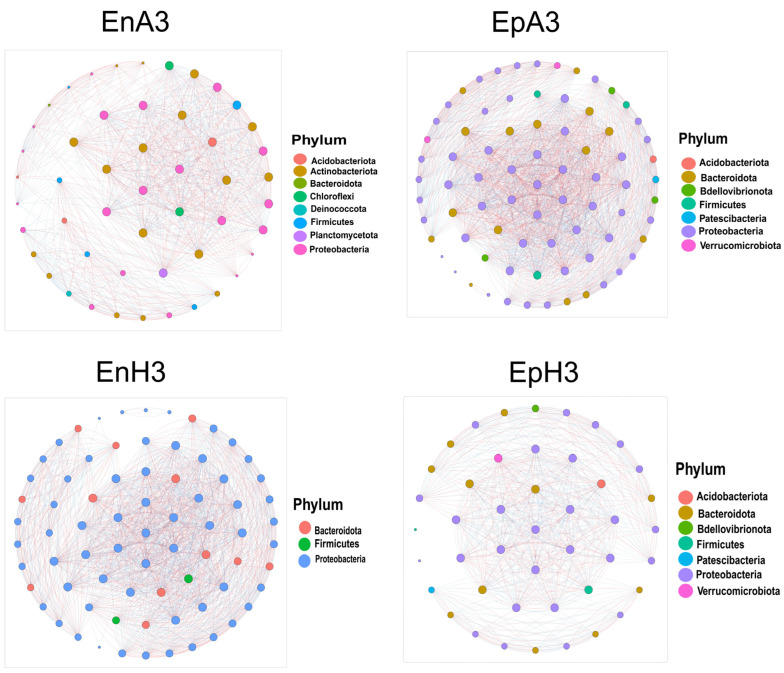
Co-occurrence networks of microbial communities: Four co-occurrence networks were constructed to analyze the relationships between microbial communities. Nodes in the networks represent individual OTUs (Operational Taxonomic Units), colored according to bacterial phyla. Edge colors indicate positive (red) and negative (blue) correlation coefficients. Spearman’s correlation coefficients (r > 0.6, *p* < 0.05) were used for network construction. The size of each node is proportional to its number of connections (degree).

**Table 1 ijms-25-13330-t001:** Topological properties of the networks for four groups.

Network Indexes	EpH3	EpA3	EnH3	EnA3
num.edges	304	942	806	446
num.pos.edges	143	502	388	233
num.neg.edges	161	440	418	213
num.vertices	42	75	74	50
connectance	0.35	0.33	0.29	0.36
average.degree	14.48	25.12	21.78	17.84
average.path.length	1	1	1	1
diameter	1	1	1	1
edge.connectivity	0	0	0	0
clustering.coefficient	1	1	1	1
no.clusters	3	4	5	3
centralization.degree	0.11	0.12	0.09	0.12
centralization.betweenness	0	0	0	0
centralization.closeness	0	0	0	0

## Data Availability

The original contributions presented in the study are included in the article; further inquiries can be directed to the corresponding author.

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
