# Peer review of "Amplicon Sequencing Analysis of Submerged Plant Microbiome Diversity and Screening for ACC Deaminase Production by Microbes"

_ijms, 2024, doi:10.3390/ijms252413330_

Round 1
Reviewer 1 Report
Comments and Suggestions for Authors
The paper investigates the microbial diversity of submerged plants, focussing on microbes that produce ACC deaminase, an enzyme that helps plants to mitigate stress. Using metagenomic analysis, the study identified key microbial genera, including Pseudomonas, Bacillus and Acinetobacter, with many strains exhibiting significant ACC deaminase activity. These microbes play a crucial role in the adaptation of plants to environmental stress and offer potential applications in sustainable agriculture.
Although the research is valuable and significant, the manuscript needs further elaboration on several important points before it is suitable for publication.
1. Abstract: We ask you to restructure the summary to make it clearer and more fluid. Please include the following elements:
An introduction to the topic
Aims of the study
Methods used
Key findings
Future prospects and wider implications of the research (this could also be emphasised at the beginning after the introduction).
2. Introduction: Please explain the concept of submerged plants. Could you also elaborate on which specific plant species are of most interest for this study?
3. Line 92: Please explain in a few sentences how the strains were isolated.
4. Line 98: Could you clarify what "DF" refers to? Please ensure that all abbreviations are introduced the first time they are used.
5. Line 100: Could you indicate where the bacteria were quantified? Was this done in the samples used for the original isolation or in a specific medium?
6. Please provide an additional figure to illustrate the data on the frequency and species distribution of ACC-positive strains (with a phylogenetic tree for the strains).
7. Line 110: Please indicate the species of the isolated strains. This will facilitate comparison with the metagenomic data presented in the following section.
8. Line 118: Could you clearly indicate which samples were used to isolate the DNA for metagenomic analysis? Was the DNA extracted from water samples, from the surface of plants or from crushed plant tissue?
9. Line 123: Please indicate which plant species were analysed.
10. Line 151: The current labelling of the samples (EnA3, EnH3, EpA3, EpH3) is unclear to the reader. Could you explain in more detail what these samples represent, how they differ and why you have chosen to show these particular samples?
11. Line 287: Please indicate the plant species in this section.
12. Line 344: Can you indicate the number of replicates used? What is the variability between the replicates?
Author Response
Comments 1: Abstract: We ask you to restructure the summary to make it clearer and more fluid. Please include the following elements:
An introduction to the topic
Aims of the study
Methods used
Key findings
Future prospects and wider implications of the research (this could also be emphasized at the beginning after the introduction
Response 1: An introduction to the topic [Submerged plants can thrive entirely underwater, playing a crucial role in maintaining water quality, supporting aquatic organisms, and enhancing sediment stability. However, they face multiple challenges, including reduced light availability, fluctuating water conditions, and limited nutrient access. Despite these stresses, submerged plants demonstrate remarkable resilience through physiological and biochemical adaptations. Additionally, their interactions with microbial communities are increasingly recognized as pivotal in mitigating these environmental stresses.] Thank you for pointing this out We agree with this comment. Therefore, we have [added an extra sentence to the abstract Page number 1, paragraph 1 and line 13-18]
Aims of the study [This research aims to identify and screen microbes from submerged plant samples capable of producing 1-aminocyclopropane-1-carboxylic acid (ACC) deaminase and to explore microbial diversity through metagenomic analysis. Microbes were isolated and screened for ACC deaminase production, and metagenomic techniques, including co-occurrence network analysis, were used to examine microbial diversity and interactions within the communities.] Thank you for pointing this out We agree with this comment. Therefore, we have [modified the following aims “This study aims to screen and identify microbes capable of producing 1-aminocyclopropane-1-carboxylic acid (ACC) deaminase from submerged plant samples and to investigate their microbial diversity through metagenome analysis” Page number 1, paragraph 1 and line 20-23]
Methods used [Microbes were isolated from submerged plant samples and screened for their ability to produce ACC deaminase. The microbial diversity was then analyzed using metagenomic techniques, with co-occurrence network analysis employed to elucidate the relationships and interactions among the microbial communities] Thank you for pointing this out We agree with this comment. Therefore, we have [added an extra sentence to the abstract Page number 1, paragraph 1 and line 23-26]
Key findings: [The microbial screening helped to identify the diverse microbes associated with ACC deaminase activity in submerged plants and Metagenome analysis paved way towards identify the impact of location in shaping the microbiome and the diversity associated with endophytic and epiphytic microbes. Co-occurrence network analysis further highlighted the intricate interactions within these microbial communities] Thank you for pointing this out We agree with this comment. Therefore, we have [added an extra sentence to the abstract Page number 1, paragraph 1 and line 36-40]
Comments 2: Introduction: Please explain the concept of submerged plants. Could you also elaborate on which specific plant species are of most interest for this study?
Response 2: [Submerged aquatic vegetation (SAV) comprises plants that grow fully underwater. These plants are vital to aquatic ecosystems, providing habitat and sustenance for diverse aquatic organisms, enhancing water quality through oxygenation, and stabilizing sediments. SAV is crucial for maintaining ecological equilibrium in both freshwater and marine environments.] Thank you for pointing this out We agree with this comment. Therefore, we have [added an extra sentence to the Introduction Page number 2, paragraph 1 and line 47-51].
[This study primarily focused on to exploring the role of native submerged plants including Myriophyllum spicatum, Najas guadalupensis, Heteranthera dubia and Ceratophyllum demersum was used widely in to understand the role of microbiome in submerged plants] Thank you for pointing this out We agree with this comment. Therefore, we have [added an extra sentence to the Introduction Page number 2, paragraph 4 and line 104-107].
Comments 3: Line 92: Please explain in a few sentences how the strains were isolated.
Response 3: [Plant samples from these locations were surface sterilize to remove the surface microbes and homogenized the isolated endophytic microbes was initially spread plated and indi-vidual colonies were streak plated Luria-Bertani (LB) media] Thank you for pointing this out We agree with this comment. Therefore, we have [added an extra sentence to the Results Page number 3, paragraph 4 and line 133-135].
Comments 4: Line 98: Could you clarify what "DF" refers to? Please ensure that all abbreviations are introduced the first time they are used.
Response 4: [Dworkin and Foster] Thank you for pointing this out We agree with this comment. Therefore, we have [added abbreviations Page number 3, paragraph 4 and line 136].
[Global Positioning System] Thank you for pointing this out We agree with this comment. Therefore, we have [added abbreviations Page number 3, paragraph 4 and line 132-133].
[Principal Coordinate Analysis] Thank you for pointing this out We agree with this comment. Therefore, we have [added abbreviations Page number 6, paragraph 2 and line 262].
[Operational Taxonomic Unit] Thank you for pointing this out We agree with this comment. Therefore, we have [added abbreviations Page number 7, paragraph 3 and line 289].
[Amplicon Sequence Variants] Thank you for pointing this out We agree with this comment. Therefore, we have [added abbreviations Page number 10, paragraph 2 and line 400].
Comments 5: Line 100: Could you indicate where the bacteria were quantified? Was this done in the samples used for the original isolation or in a specific medium?
Response 5: [The isolated microbes were screened in DF (Dworkin and Foster) + ACC media and a total of 177 endophytic microbial strains were isolated, and molecular identification was per-formed on these samples] Thank you for pointing this out We agree with this comment. Therefore, we have [modified the sentence in the Results Page number 3, paragraph 4 and line 136-137].
Comments 6: Please provide an additional figure to illustrate the data on the frequency and species distribution of ACC-positive strains (with a phylogenetic tree for the strains).
Response 6: Additional figure containing a phylogenetic tree including the screened microbes are included in the supplementary figures.
Comments 7: Line 110: Please indicate the species of the isolated strains. This will facilitate comparison with the metagenomic data presented in the following section.
Response 7: [Plant samples collected from location A and H was Myriophyllum spicatum and Najas minor respectively. The isolation of the DNA for Amplicon Sequencing analysis was done from the shoot samples collected from location A and H and both epiphytic and en-dophytic sample DNA isolation was performed] Thank you for pointing this out We agree with this comment. Therefore, we have [added an extra sentence in the Results Page number 4, paragraph 2 and line 163-166].
Comments 8: Line 118: Could you clearly indicate which samples were used to isolate the DNA for metagenomic analysis? Was the DNA extracted from water samples, from the surface of plants or from crushed plant tissue?
Response 8: [The isolation of the DNA for metagenome analysis was done from the shoot samples collected from location A and H and both epiphytic and endophytic sample DNA isolation was performed] Thank you for pointing this out We agree with this comment. Therefore, we have [added an extra sentence in the Results Page number 4, paragraph 2 and line 163-165].
Comments 9: Line 123: Please indicate which plant species were analysed.
Response 9: [Plant samples collected from location A and H was Myriophyllum spicatum and Najas minor respectively] Thank you for pointing this out We agree with this comment. Therefore, we have [added an extra sentence in the Results Page number 4, paragraph 2 and line 162-163].
Comments 10: Line 151: The current labelling of the samples (EnA3, EnH3, EpA3, EpH3) is unclear to the reader. Could you explain in more detail what these samples represent, how they differ and why you have chosen to show these particular samples?
Response 10: [Bar graphs illustrating the bacterial community compositions at different taxonomic levels across four samples (Endophytic A3 (EnA3), Endophytic H3 (EnH3), Epiphytic A3 (EpA3), Epiphytic H3 (EpH3)): (a) Phylum level, (b) Family level, and (c) Genus level, depicting the shifts in microbial diversity and abundance] Thank you for pointing this out We agree with this comment. Therefore, we have [modified the sentence in the Results Page number 5, paragraph 1 and line 200-201].
Comments 11: Line 287: Please indicate the plant species in this section.
Response 11: [This study primarily focused on to exploring the role of submerged plants found along Alabama including Myriophyllum spicatum, Najas guadalupensis, Heteranthera dubia and Ceratophyllum demersum which was used widely in our experiment to screen microbes] Thank you for pointing this out We agree with this comment. Therefore, we have [added an extra sentence in the Results Page number 8, paragraph 5 and line 339-342].
Comments 12: Line 344: Can you indicate the number of replicates used? What is the variability between the replicates?
Response 12: [ The metagenome data analysis were carried out using 3 bioreps for each samples studies and variability in the study is represented in the Table S4] Thank you for pointing this out We agree with this comment. Therefore, we have [added an extra sentence in the Data analysis Page number 10, paragraph 2 and line 399-400].
Reviewer 2 Report
Comments and Suggestions for Authors
1 The summary lacks a concluding section. 2 Keywords should not be the same as the title. 3 The article mentions the risks faced by aquatic ecosystems in the introduction, but lacks a detailed discussion of the specific sources and impacts of these risks. It is recommended to add detailed background information on how factors such as floods and pollution affect aquatic ecosystems and plant microbiomes. 4 However, the purpose of the study is somewhat vaguely stated in the introduction. It is recommended to clearly state the potential significance of this study for agriculture, ecological restoration, or environmental management. 5 The article presents a large amount of data and charts in the results section, but the interpretation and discussion of these data are relatively insufficient. It is recommended to strengthen the analysis of the results, such as explaining the differences in microbial diversity between different samples, changes in ACC deaminase activity levels, and their potential causes. 6 The discussion section should delve deeper into the significance and potential impacts of the study's findings. For example, the specific mechanisms by which ACC deaminase-producing microorganisms enhance plant stress resistance and productivity can be discussed, as well as the potential applications of these findings for agricultural practices. 7 The article refers to previous studies in several sections, but the citation and comparison of these studies are relatively limited. It is recommended to increase the citation of other relevant studies and discuss in detail the connections and differences between these studies and the results of this paper. 8 The figures in the article could be clearer. 9 The conclusion section should succinctly summarize the main findings and significance of the article. It is recommended to remove redundant expressions and emphasize the innovative aspects and potential application value of the research.
Author Response
Comments 1: The summary lacks a concluding section
Response 1: [The microbial screening helped to identify the diverse microbes associated with ACC deaminase activity in submerged plants and Metagenome analysis paved way towards identify the impact of location in shaping the microbiome and the diversity associated with endophytic and epiphytic microbes. Co-occurrence network analysis further highlighted the intricate interactions within these microbial communities] Thank you for pointing this out We agree with this comment. The review comment was also raised by the reviewer 1 hence a common modification was added for the response. Therefore, we have [added an extra sentence to the abstract Page number 1, paragraph 1 and line 36-40]
Comments 2: Keywords should not be the same as the title
Response 2: [Aquatic plants, Stress tolerance, Co-occurrence network analysis, Natural sources] Thank you for pointing this out We agree with this comment. Therefore, we have [modified the following keywords “Microbiome, ACC deaminase, Metagenome, Plant-microbe interaction, Co-occurrence network” Page number 1, paragraph 2 and line 44]
Comments 3: The article mentions the risks faced by aquatic ecosystems in the introduction but lacks a detailed discussion of the specific sources and impacts of these risks. It is recommended to add detailed background information on how factors such as floods and pollution affect aquatic ecosystems and plant microbiomes.
Response 3: [Aquatic ecosystems face numerous critical threats that threatens plant microbiomes and the overall health of the ecosystem. Flooding and extreme weather events introduce sediment, debris, and pollutants into aquatic ecosystems, potentially suffocating aquatic life. Nutrient pollution from agricultural runoff and sewage can cause eutrophication and hypoxia, while heavy metals and organic pollutants can accumulate and become toxic. Habitat destruction due to development and the introduction of invasive species can disrupt food webs and decrease biodiversity. Climate change is worsening many of these issues, causing shifts in species distributions and the collapse of sensitive environments. Addressing these complex risks is essential to protect the health of aquatic ecosystems and the microbial communities they sustain] Thank you for pointing this out We agree with this comment. Therefore, we have [added an extra sentence to the introduction Page number 2, paragraph 1 and line 53-63]
Comments 4: However, the purpose of the study is somewhat vaguely stated in the introduction. It is recommended to clearly state the potential significance of this study for agriculture, ecological restoration, or environmental management
Response 4: [Research on the microbiomes associated with submerged aquatic plants is crucial for tackling significant environmental challenges. By deepening our understanding of the diversity and functions of these microbial communities, we can discover innovative applications for practical benefits. In agriculture, utilizing beneficial microbes, such as those producing ACC deaminase, could promote more sustainable and climate-resilient crop-ping systems. For ecological restoration, knowledge of how microbiomes support aquatic ecosystem health can inform efforts to rehabilitate degraded habitats. From an environ-mental management standpoint, studying the complex microbial interactions that drive nutrient cycling and ecosystem functions can shape policies to protect sensitive aquatic environments. Overall, this research on submerged plant microbiomes is vital for advancing sustainable agricultural practices, restoring degraded ecosystems, and enhancing environmental restoration efforts] Thank you for pointing this out We agree with this comment. Therefore, we have [added an extra sentence to the introduction Page number 3, paragraph 3 and line 117-128]
Comments 5: The article presents a large amount of data and charts in the results section, but the interpretation and discussion of these data are relatively insufficient. It is recommended to strengthen the analysis of the results, such as explaining the differences in microbial diversity between different samples, changes in ACC deaminase activity levels, and their potential causes.
Response 5: [Additionally, there was a significant variation in ACC deaminase activity levels among the isolated microbial strains. While some strains showed strong growth and viability in the presence of ACC, others had noticeably lower activity[17,18]. This finding is consistent with earlier research, which indicates that the variability in ACC deaminase activity among different strains can be attributed to factors such as their taxonomic classification, metabolic pathways, and environmental conditions[19,20]. Further investigation into these factors could provide valuable insights into how these microbes enhance plant resilience to flooding stress.] Thank you for pointing this out We agree with this comment. Therefore, we have [added an extra sentence to the discussion Page number 8, paragraph 1 and line 302-310]
[A significant observation was the difference in microbial diversity between endophytic and epiphytic samples, as well as across various sampling sites. The taxonomic analysis at the family and genus levels (Figures 1b and 1c) highlighted distinct microbial commu-nity compositions. Endophytic samples showed higher overall diversity, with notable representation of Sphingomonadaceae and Moraxellaceae. In contrast, epiphytic samples were dominated by Rhodocyclaceae, Pseudomonadaceae, and Oxalobacteraceae. These findings are consistent with previous studies, which have indicated that epiphytic com-munities are generally more diverse and less stable than endophytic ones[30]. Moreover, plant identity has been shown to have a more pronounced impact on the diversity and structure of epiphytic bacteria compared to endophytic bacteria[31]. Additionally, re-search on the grapevine leaf microbiome has revealed that epiphytic isolates are phyloge-netically more diverse than endophytic isolates[32].] Thank you for pointing this out We agree with this comment. Therefore, we have [added an extra sentence to the discussion Page number 8, paragraph 4 and line 343-355].
Comments 6: The discussion section should delve deeper into the significance and potential impacts of the study's findings. For example, the specific mechanisms by which ACC deaminase-producing microorganisms enhance plant stress resistance and productivity can be discussed, as well as the potential applications of these findings for agricultural practices.
Response 6: [The identification of 50 microbial strains capable of producing ACC deaminase from submerged plant microbiomes shows great potential for improving plant stress tolerance and promoting sustainable agriculture. These microbes utilize a key mechanism by breaking down ACC (a precursor of ethylene) into α-ketobutyrate and ammonia, which helps lower ethylene levels in plants experiencing flooding stress [21]. This reduction in ethylene is crucial for maintaining normal root development, enhancing water and nutri-ent uptake, and boosting overall plant resilience under waterlogged conditions [18,22]. The high prevalence of these beneficial microbes in our samples indicates a strong potential for developing targeted bioinoculants to help crops better withstand flooding stress. This approach is particularly relevant given the increasing frequency of extreme weather events due to climate change, offering a sustainable method to enhance agricultural productivity in flood-prone areas while reducing reliance on chemical inputs.] Thank you for pointing this out We agree with this comment. Therefore, we have [added an extra sentence to the discussion Page number 8, paragraph 2 and line 311-323].
Comments 7: The article refers to previous studies in several sections, but the citation and comparison of these studies are relatively limited. It is recommended to increase the citation of other relevant studies and discuss in detail the connections and differences between these studies and the results of this paper
Response 7: [Shahid, M., Singh, U.B., Khan, M.S., Singh, P., Kumar, R., Singh, R.N., Kumar, A. and Singh, H.V., 2023. Bacterial ACC deaminase: Insights into enzymology, biochemistry, genetics, and potential role in amelioration of environmental stress in crop plants. Frontiers in microbiology, 14, p.1132770.] Thank you for pointing this out We agree with this comment. Therefore, we have [added citation to the discussion Page number 8, paragraph 1 and line 306].
[Shi, Y., Yuan, Y., Feng, Y., Zhang, Y. and Fan, Y., 2023. Bacterial Diversity Analysis and Screening for ACC Deami-nase-Producing Strains in Moss-Covered Soil at Different Altitudes in Tianshan Mountains—A Case Study of Glacier No. 1. Microorganisms, 11(6), p.1521.] Thank you for pointing this out We agree with this comment. Therefore, we have [added citation to the discussion Page number 8, paragraph 1 and line 306].
[Verma, P.P., Sharma, S.G. and Kaur, M., 2020. Microbial ACC-Deaminase Attributes: Perspectives and Applications in Stress Agriculture. Advances in Plant Microbiome and Sustainable Agriculture: Functional Annotation and Future Challenges, pp.65-83.] Thank you for pointing this out We agree with this comment. Therefore, we have [added citation to the discussion Page number 8, paragraph 1 and line 309].
[Gupta, A., Rai, S., Bano, A., Sharma, S., Kumar, M., Binsuwaidan, R., Suhail Khan, M., Upadhyay, T.K., Alshammari, N., Saeed, M. and Pathak, N., 2022. ACC deaminase produced by PGPR mitigates the adverse effect of osmotic and salinity stresses in Pisum sativum through modulating the antioxidants activities. Plants, 11(24), p.3419.] Thank you for pointing this out We agree with this comment. Therefore, we have [added citation to the discussion Page number 8, paragraph 1 and line 309].
[Gupta, S. and Pandey, S., 2019. ACC deaminase producing bacteria with multifarious plant growth promoting traits al-leviates salinity stress in French bean (Phaseolus vulgaris) plants. Frontiers in microbiology, 10, p.1506.] Thank you for pointing this out We agree with this comment. Therefore, we have [added citation to the discussion Page number 8, paragraph 2 and line 315].
[Gupta, A., Bano, A., Rai, S., Kumar, M., Ali, J., Sharma, S. and Pathak, N., 2021. ACC deaminase producing plant growth promoting rhizobacteria enhance salinity stress tolerance in Pisum sativum. 3 Biotech, 11(12), p.514.] Thank you for pointing this out We agree with this comment. Therefore, we have [added citation to the discussion Page number 8, paragraph 2 and line 317].
[Sanjenbam, P. and Agashe, D., 2024. Divergence and convergence in epiphytic and endophytic phyllosphere bacterial communities of rice landraces. mSphere, pp.e00765-24.] Thank you for pointing this out We agree with this comment. Therefore, we have [added citation to the discussion Page number 9, paragraph 1 and line 363].
[Yao, H., Sun, X., He, C., Li, X.C. and Guo, L.D., 2020. Host identity is more important in structuring bacterial epiphytes than endophytes in a tropical mangrove forest. FEMS microbiology ecology, 96(4), p.fiaa038.] Thank you for pointing this out We agree with this comment. Therefore, we have [added citation to the discussion Page number 9, paragraph 1 and line 365].
[Bruisson, S., Zufferey, M., L’Haridon, F., Trutmann, E., Anand, A., Dutartre, A., De Vrieze, M. and Weisskopf, L., 2019. Endophytes and epiphytes from the grapevine leaf microbiome as potential biocontrol agents against phytopathogens. Frontiers in microbiology, 10, p.2726] Thank you for pointing this out We agree with this comment. Therefore, we have [added citation to the discussion Page number 9, paragraph 1 and line 366].
Comments 8: The figures in the article could be clearer.
Response 8: Thank you for pointing this out We agree with this comment. Therefore, we have added high quality images to the revised paper
Reviewer 3 Report
Comments and Suggestions for Authors
I have carefully reviewed the manuscript entitled “Metagenome analysis of Submerged plant microbiome diversity and screening for ACC deaminase production microbes” by Mohan et al. to a MDPI journal International Journal of Molecular Sciences. This study aims to screen and identify microbes capable of producing ACC deaminase from submerged plant samples and to investigate their microbial diversity through metagenome analysis. Experimental design is reasonable, and the testing methods and statistical techniques used here are also correct. Although this study is very simple, the research results can benefit to advance the understanding of the interactions between plants, microorganisms and the environment to a certain extent. I suggest rejecting this manuscript and giving it a chance to resubmit after a thorough revision. Some comments are as the following:
1. In Figure 1, statistical testing should be supplemented to more clearly reflect which taxa have significant differences in average relative abundance between four groups of samples.
2. Table 1 is missed.
3. According the results represented in the manuscript, this study only applied high-throughput amplicon sequencing of bacteria instead of metagenomics. There is a significant difference between these two approaches. Authors should not exaggerate the method used.
4. Line 340-343. Please provide a clear description involved in amplicon sequencing. Which region of 16S rRNA gene is amplified, V3-V4 or others. What are PCR reaction mixture and thermal cycle conditions and primer sets?
5. For the obtained strains capable of producing ACC deaminase, their phylogenetic trees need to be constructed and added to the manuscript.
6. The relationships among average relative abundances of the major bacterial taxa, plant growth and development and metabolism and biomass, and environmental factors should be tested, because these are very important for advancing the understanding of the interactions between plant and microorganisms and environments.
7. The Materials and Methods section and the Discussion section are too simply, particularly the latter, which cannot fully explain the results obtained from this study.
8. A clear research conclusion should be provided in the abstract section.
Author Response
Comments 1: In Figure 1, statistical testing should be supplemented to more clearly reflect which taxa have significant differences in average relative abundance between four groups of samples.
Response 1: Aqsa Thank you for pointing this out We agree with this comment. Therefore, we have added a supplementary Table S4 covering statistical significance of the result
Comments 2: Table 1 is missed.
Response 2: [Submerged plants were collected from 15 locations within the Greater Birmingham Area in Alabama, USA, as listed in Table S1.] Thank you for pointing this out We agree with this comment. Therefore, we have [modified the following sentence “Submerged plants were collected from 15 locations within the Greater Birmingham Area in Alabama, USA, as listed in Table 1.” Page number 9, paragraph 2 and line 372]
Comments 3: According to the results represented in the manuscript, this study only applied high-throughput amplicon sequencing of bacteria instead of metagenomics. There is a significant difference between these two approaches. Authors should not exaggerate the method used.
Response 3: [Amplicon Sequencing] Thank you for pointing this out We agree with this comment. Therefore, we have [changes metagenome to amplicon sequencing throughout the paper Page number 1, paragraph 1 and line 2].
Comments 4: Line 340-343. Please provide a clear description involved in amplicon sequencing. Which region of 16S rRNA gene is amplified, V3-V4 or others. What are PCR reaction mixture and thermal cycle conditions and primer sets?
Response 4: [For DNA amplification, we targeted the V4 region of the 16S rRNA gene sequence. The forward primer sequence was 515F-Y (5′-GTGYCAGCMGCCGCGGTAA-3′) paired with the reverse primer 806R (5′-GGACTACNVGGGTWTCTAAT-3′). The am-plification process utilized Phusion High-Fidelity DNA Polymerase following the pro-tocol provided by the supplier] Thank you for pointing this out We agree with this comment. Therefore, we have [modified the following sentence in Materials and methods Page number 10, paragraph 4 and line 428-432]
Comments 5: For the obtained strains capable of producing ACC deaminase, their phylogenetic trees need to be constructed and added to the manuscript.
Response 5: Additional figure containing a phylogenetic tree including the screened microbes are included in the supplementary figures S2
Comments 6: The relationships among average relative abundances of the major bacterial taxa, plant growth and development and metabolism and biomass, and environmental factors should be tested, because these are very important for advancing the understanding of the interactions between plant and microorganisms and environments.
Response 6: Understanding the relationships among the average relative abundances of major bacterial taxa, plant growth, development, metabolism, biomass, and environmental factors is indeed crucial. However, we did not collect data on environmental factors, which limited our ability to perform such analyses. We appreciate the reviewer’s insightful comment and for highlighting this important aspect for discussion.
Comments 7: The Materials and Methods section and the Discussion section are too simply, particularly the latter, which cannot fully explain the results obtained from this study.
Response 7:
Comments 8: A clear research conclusion should be provided in the abstract section.
Ressponse 8: [The microbial screening helped to identify the diverse microbes associated with ACC deaminase activity in submerged plants and Metagenome analysis paved way towards identify the impact of location in shaping the microbiome and the diversity associated with endophytic and epiphytic microbes. Co-occurrence network analysis further highlighted the intricate interactions within these microbial communities] Thank you for pointing this out We agree with this comment. The review comment was also raised other reviewers hence a common modification was added for the response. Therefore, we have [added an extra sentence to the abstract Page number 1, paragraph 1 and line 36-40]
Round 2
Reviewer 2 Report
Comments and Suggestions for Authors
good revisions
Reviewer 3 Report
Comments and Suggestions for Authors
All problems raised previously have been well addressed. I suggest accepting this manuscript for publication.